# SARS-CoV-2 in severe COVID-19 induces a TGF-β-dominated chronic immune response that does not target itself

Marta Ferreira-Gomes [1,19], Andrey Kruglov[1,2,3,19], Pawel Durek[1,19], Frederik Heinrich[1,19], Caroline Tizian[4,5,6,19], Gitta Anne Heinz [1,19], Anna Pascual-Reguant [1,7], Weijie Du[1], Ronja Mothes[1,8], Chaofan Fan[1], Stefan Frischbutter [9], Katharina Habenicht[10], Lisa Budzinski [1], Justus Ninnemann[1], Peter K. Jani[1], Gabriela Maria Guerra[1], Katrin Lehmann[1], Mareen Matz[11], Lennard Ostendorf [1,8], Lukas Heiberger[1], Hyun-Dong Chang [1,12], Sandy Bauherr [1], Marcus Maurer [9], Günther Schönrich[13], Martin Raftery[13], Tilmann Kallinich[1,5,14], Marcus Alexander Mall [5,14,15], Stefan Angermair [16], Sascha Treskatsch[16], Thomas Dörner[1,7], Victor Max Corman [13], Andreas Diefenbach [4,5,6], Hans-Dieter Volk [11,17], Sefer Elezkurtaj[18], Thomas H. Winkler [10], Jun Dong[1], Anja Erika Hauser [1,7], Helena Radbruch [8,20], Mario Witkowski[4,5,6,20], Fritz Melchers[1,20], Andreas Radbruch [1,20] & Mir-Farzin Mashreghi [1,11,20✉]

The pathogenesis of severe COVID-19 reflects an inefficient immune reaction to SARS-CoV-2. Here we analyze, at the single cell level, plasmablasts egressed into the blood to study the dynamics of adaptive immune response in COVID-19 patients requiring intensive care. Before seroconversion in response to SARS-CoV-2 spike protein, peripheral plasmablasts display a type 1 interferon-induced gene expression signature; however, following seroconversion, plasmablasts lose this signature, express instead gene signatures induced by IL-21 and TGF-β, and produce mostly IgG1 and IgA1. In the sustained immune reaction from COVID-19 patients, plasmablasts shift to the expression of IgA2, thereby reflecting an instruction by TGF-β. Despite their continued presence in the blood, plasmablasts are not found in the lungs of deceased COVID-19 patients, nor does patient IgA2 binds to the dominant antigens of SARS-CoV-2. Our results thus suggest that, in severe COVID-19, SARS-CoV-2 triggers a chronic immune reaction that is instructed by TGF-β, and is distracted from itself.

A full list of author affiliations appears at the end of the paper.

One of the properties of SARS-CoV-2 which makes it spread so successfully, is the heterogeneity of human immune responses it triggers—from asymptomatic to fatal[1]. However, the molecular reasons for this heterogeneity of COVID-19 are not clear. A predisposition for severe COVID-19 is less likely to be just related to the dose of pathogen or to a distinct route of infection, but rather to advanced age, suggesting that inefficient immune responses may play a pivotal role[2]. In the response to airborne viral pathogens, antibodies are believed to play a critical role in preventing docking onto the host cells, but also in preventing systemic spread[3]. While secretory IgA antibodies protect mucosal surfaces, IgG antibodies provide systemical protection. Antibodies are generated in immune reactions, mostly in secondary lymphoid organs, when activated B lymphocytes are instructed to differentiate into antibody-secreting plasmablasts. These can then leave the secondary lymphoid organs, migrate to the bone marrow and there become (longlived) plasma cells[4]. On their way to the bone marrow, plasmablasts can be detected in the blood starting around 7 days after onset of an acute immune response, and only for 1–2 days[5]. In antibody-mediated autoimmune diseases, like systemic lupus erythematosus, their persistence in the blood indicates chronic immune reactions and continued activation of B lymphocytes[6].

In a certain sense, plasmablasts are perfect reporters of the immune reaction, since their transcriptome reflects instruction by antigen, T lymphocytes and other cues from their environment. Particularly, interaction of activated B lymphocytes with CD4 T helper lymphocytes is essential for the generation of immunological memory, for affinity maturation by somatic hypermutation, and it controls antibody class switching[7]. Antibody-class switching to distinct antibody classes (isotypes), providing the antibodies with defined functional properties, is controlled by distinct cytokines which can be provided by the T helper lymphocytes, but also could be derived from other sources. In particular, class switch recombination is targeted to IgG1 by interleukin-21[8,9], IgG2 by interferon-γ[10], and IgA1 and IgA2 by TGF-β[11]. TGF-β is also known as a prominent regulator of immune reactions[12], and it causes fibrosis[13], a comorbidity of severe COVID-19 patients[14,15].

Here we show, for a cohort of patients with severe COVID-19 requiring intensive medical care for up to 60 days, the continuous egress of plasmablasts into the blood, reflecting a continued immune reaction. Initially, within the first week of ICU admission, this immune reaction is directed against SARS-CoV-2, as all patients acquired IgG antibodies against the S and N proteins. Later, IgA-expressing plasmablasts are generated predominantly, reflecting continued instruction of the B lymphocytes by TGF-β. However, regarding their specificity, only one-third of the patients express S protein-specific IgA, and only 1 of those expressed S protein-specific IgA2, the terminal antibody class targeted by TGF-β. Circulating plasmablasts are clonally expanded and their antibodies are somatically mutated, but not specific for SARS-CoV-2 S or N protein. Taken together, these results point to TGF-β as a key cytokine regulating a chronic immune reaction in severe COVID-19, an immune reaction which is no longer directed to SARS-CoV-2.

## Results

**COVID-19 plasmablast transcriptomes over time**. To analyze the adaptive immune response triggered by SARS-CoV-2 in severely affected COVID-19 patients, we initially focused on plasmablasts of the peripheral blood, considering them as biosensors of the ongoing immune reaction and effectors of humoral immunity[5,16]. To do so, we established a cohort (characterized in Supplementary Table 1) of 11 COVID-19 intensive care unit

(ICU) patients, admitted to the ICU according to the QuickSOFA score of clinical parameters, the POST-SAVE-Concept of the state of Berlin[17]. We analyzed samples from six patients within the first week after ICU admission, all except from one (patient #11, day 7) being seronegative for the spike (S) protein of SARS-CoV-2, and from seven patients who had been in the ICU for more than a week (9–59 days) and who were seropositive (IgG) for SARS-CoV-2 S protein. Note that three of the patients were analyzed both at early and late time points (patient #1, 3, 8). Three healthy controls were also included. CD27$^{high}$CD38$^{high}$ cells of the B cell lineage were enriched from peripheral blood by MACS and FACS (Supplementary Fig. 1a). The frequencies of CD27$^{high}$CD38$^{high}$ B cells were significantly elevated in the peripheral blood of COVID-19 patients compared to healthy controls (Fig. 1a). From the enriched cells we generated single cell transcriptomes and B cell receptor (BCR) repertoires. According to their transcriptomes, cells were clustered by Uniform Manifold Approximation and Projection for Dimension Reduction (UMAP)[18]. UMAP defined six distinct clusters of CD27$^{high}$CD38$^{high}$ B cells in the patients and healthy controls analyzed (Fig. 1b–d), although not all patients and controls showed cells in all clusters (Fig. 1e, f, Supplementary Fig. 1b). Cluster 1 contained predominantly CD19$^+$CD20$^+$CD27$^+$ experienced B lymphocytes, with a gene expression signature resembling that of resting B lymphocytes[19] (Fig. 1b, d). Cells of cluster 6 were IRF4$^{high}$PRDM1$^{high}$ late plasmablasts and plasma cells[20] (Fig. 1b, d). Cells of clusters 2 and 3 represented CD19$^+$CD27$^+$CD38$^+$ plasmablasts, with cells of cluster 2 expressing MIKI67, i.e. proliferating, and cells of cluster 3 not expressing MKI67, marking them as cells in proliferative rest (Fig. 1b, d). Cells of clusters 4 and 5 were also CD19$^+$CD27$^+$CD38$^+$ plasmablasts, but distinct from those of clusters 2 and 3 by expression of interferon (IFN) signature genes, like IFIT1[21] (Fig. 1b, d). While cells of cluster 4 did express MKI67, cells of cluster 5 did not. Of note, cells of clusters 2 and 4 as well as 3 and 5 are transcriptionally related (Fig. 1c). Cells of clusters 2–5 lost expression of MS4A1 (CD20) but expressed IRF4 and PRDM1, as well as CD40, HLA-DR, IGJ, and integrins and chemokine receptors important for migration into mucosal tissue[22] (Fig. 1d, Supplementary Fig. 1c), thus characterizing them as plasmablasts on their way to differentiate into antibody-secreting plasma cells. Cells of clusters 1–3 were dominant in healthy individuals, who showed very few circulating plasma cells (cluster 6) and no plasmablasts expressing IFN-induced signature genes (cluster 5) (Fig. 1e, f). Cells expressing IFN-induced signature genes were the most abundant in the six "early" ICU COVID-19 patients, who also showed a prominent population of plasma cells (cluster 6), but only very few experienced, resting B cells (cluster 1). In the eight patients who had been in the ICU for more than 7 days (including three who were also analyzed at early time points), plasmablasts with an IFN-induced gene signature were not present (Fig. 1e, f). Instead, prominent populations of clusters 2 and 3 became apparent. Circulating plasma cells were still abundant, while only a few experienced B cells were captured and sequenced (Fig. 1e, f). Taken together, patients recently admitted to the ICU showed a prominent egress of IFN-driven plasmablasts and plasma cells into the peripheral blood, while in patients with prolonged ICU stays (more than 7 days), circulating plasmablasts and plasma cells were still prominent, but the underlying immune reaction was no longer driven by IFNs.

**Severe COVID-19 is instructed by TGF-β**. All patients who had been in the ICU for more than 7 days had in their serum SARS-CoV-2 spike (S) protein-specific IgM, and IgG antibodies. Only four of them had S-specific IgA1 antibodies and only one showed detectable titers of S-specific IgA2 antibodies (Fig. 2a,

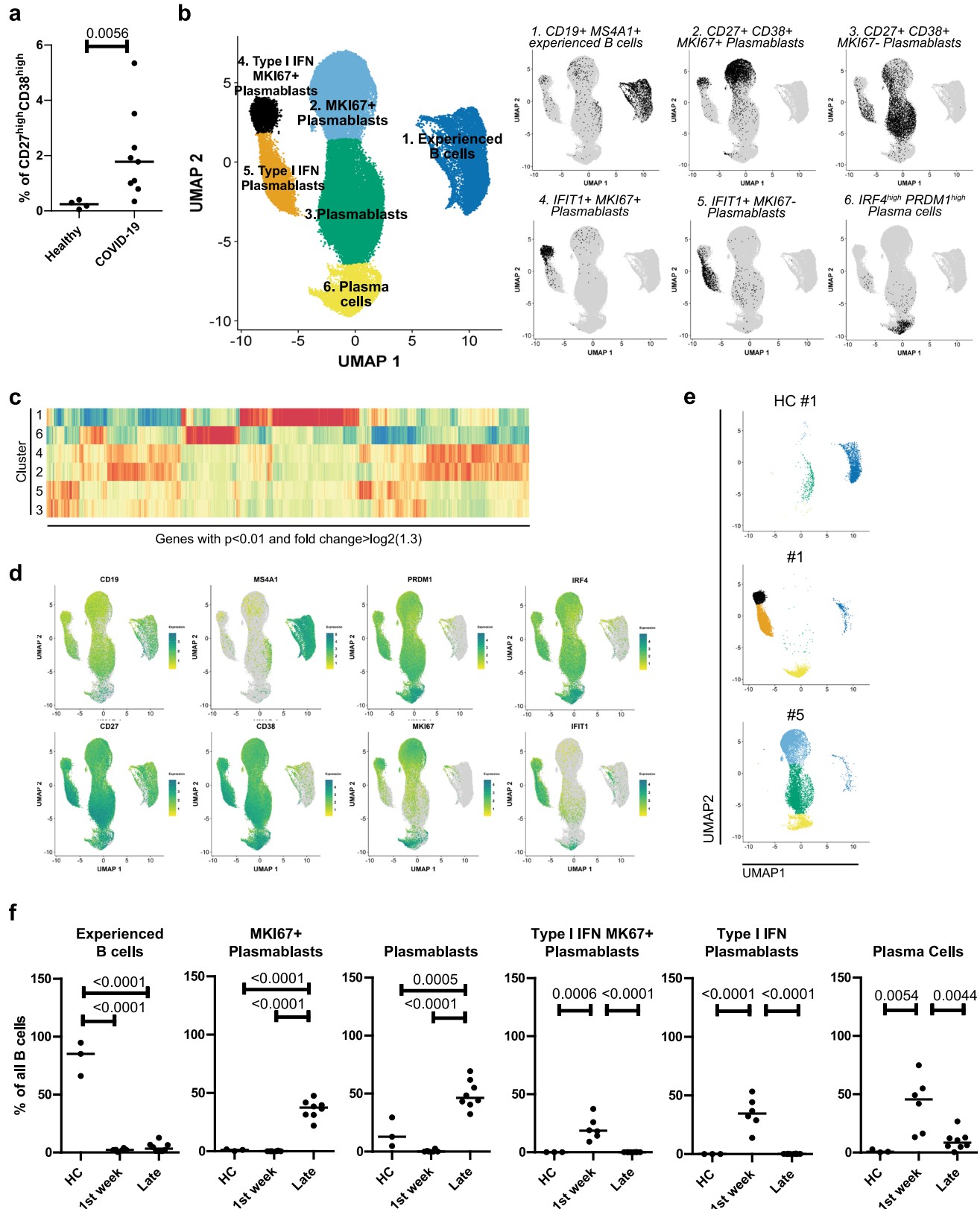

Supplementary Fig. 2a–d). Specific antibodies were measured both by ELISA and by cytometric serology using human embryonic kidney (HEK) cells transformed to express SARS-CoV-2 S protein. Both assays correlated significantly (Supplementary Fig. 2e). This indicated that SARS-CoV-2 had induced a significant humoral immune reaction, including the activation of SARS-CoV-2-specific B cells, antibody class switching to IgG and IgA, and their differentiation into plasma cells. Antibody classes characterize the immune reaction underlying the egress of plasmablasts and plasma cells into the blood. Plasmablasts of COVID-19 patients expressed most prominently IgM, IgG1, IgA1, IgG2, and IgA2 antibodies (Fig. 2b). Instruction to switch to

**Fig. 1 Single cell transcriptomes of COVID-19 patients show dynamic differences of activated B cell subpopulations at different stages after ICU admission.** Peripheral blood activated/differentiated B cells (CD27$^{high}$ CD38$^{high}$) were isolated and sorted by FACS for single cell sequencing (gating strategy in Supplementary Fig. 1a). **a** Percentage of CD27$^{high}$ CD38$^{high}$ cells among live B cells. Each sample is represented by a circle. The line indicates the median. Unpaired two-sided Mann–Whitney $U$ test. **b** UMAP representation of 72,277 CD27$^{high}$ CD38$^{high}$ sorted B cells from 9 COVID-19 ICU patients and three healthy controls. Between 2416 and 11,229 cells were recovered per sample. Transcriptionally similar clusters were identified using shared nearest neighbor (SNN) modularity optimization (left). The combination of *CD19, MS4A1, CD27, CD38, IFIT1, MIKI67, IRF4,* and *PRDM1* expression were used for annotation of the different activation/differentiation stages (right). **c** Heatmap of genes that resemble markers for the six different clusters. Depicted are genes with a *p*-value < 0.01 (two-sided Wilcoxon rank sum test) after Bonferroni correction, and an average absolute fold-change > log2(1.3). Shown are *z*-scores of the average expression. **d** UMAP representation of the expression levels of selected signature genes for activated/differentiated B cells. **e** UMAP representation of analyzed cells from one healthy control and two ICU patients, representing early (first week, patient #1) and late (>7days, patient #5) phase after ICU admission. **f** Percentage of cells belonging to a defined cluster among all sequenced B cells per donor. Each dot represents one time point from one single donor. Donors were grouped as "HC" for healthy controls (3 donors), "1st week" for patients within 7 days after ICU admission (six donors) and "Late" for patients who have been admitted to the ICU for more than a week at the time of analysis (eight donors). The line indicates the median. Significance was determined by using a two-sided analysis of varience (ANOVA) folowed by Tukey's multiple comparison test with corrected p values as indicated in the figure.

IgG1 is controlled by the cytokines IL-10 and IL-21[8,9], while switching to IgG2 is controlled by IFN-γ[10], and switching to IgA1 and IgA2 by TGF-β[11]. Considering the arrangement of the genes for antibody heavy chain constant regions on the chromosome[23], sequential switching from IgG1 to IgA is possible, also from IgA1 to IgG2, but not the reverse, IgA2 being the 3′ terminal gene for the antibody heavy chain locus. Plasmablasts expressing IgM, i.e. not yet class switched, were most frequent in healthy controls and in the COVID-19 patients early at ICU, at frequencies of up to 18% (Fig. 2c). In patients hospitalized in the ICU for longer, IgM+ cells were below or around 5%. Plasmablasts expressing IgG2, reflecting switch instruction by IFNs, were most frequent in patient #2, i.e. early on in ICU and before seroconversion, and in patient #8. Plasmablasts expressing IgG1 were rare (2%) among healthy donors, but frequent in COVID-19 patients, ranging from 17% to 38%. Interestingly, in all patients except patient #7, these IgG1+ cells define coherent subpopulations within clusters 2–5, which also contain the IgA1-expressing cells. The third most abundant antibody class found in the COVID-19 patients was IgA2. IgA2-expressing cells formed a subpopulation within cluster 3, different from the IgG1/IgA1-expressing cells in patients #3–#9. These data allow us to conclude that in early stages of severe COVID-19 the switch instruction of IFN-instructed B cells (clusters 4 and 5) had been dominated by IL-21 and TGF-β, leading most cells to express IgG1 or IgA1. In prolonged ICU stay, switch instruction by TGF-β appears to be continuing, inducing the cells to switch towards the most distal of all antibody classes, IgA2, resulting in a significant ($p = 0.008$) increase in the frequency of IgA2-expressing plasmablasts when comparing patients early and late in ICU (Fig. 2b). Even on a systemic level, serum concentrations of IgA2 increase significantly over time in ICU, while IgG and IgA1 levels do not (Fig. 2d, Supplementary Fig. 2f). Continued switch instruction by TGF-β is also evident from the expression of Iα1 and Iα2 switch transcripts in the plasmablasts of clusters 2 and 3 of patients #3–#8 (Supplementary Fig. 3a, b). These switch transcripts reflect switch instruction and precede switch recombination, both on the active and on the allelically excluded Ig heavy chain locus[24,25]. The impact of prolonged TGF-β signaling on plasmablasts of cluster 3 is not only evident from the switch transcripts, but also from gene sets obtained from previously published data on human IL-2/IL-21-activated B cells (expanded for 6 days for differentiation into plasmablasts) aditionally treated for different time periods with TGF-β in the presence of IL-6 and IL-21[26]. While plasmablasts of clusters 2 and 4 were enriched for genes of IL-2/IL-21-activated B cells (Supplementary Fig. 3c), cells of clusters 4 and 5 showed enrichment for genes which are expressed in IL-2/IL-21-activated B cells in the presence of IL-6/IL-21 and IFN-α for 12 h

(Supplementary Fig. 3c). Plasmablasts of cluster 3 obtained from patients #3 to #6 are enriched for genes that are characteristic for IL-2/IL-21-activated B cells that have been stimulated with IL-6/IL-21 and TGF-β for 12 h, and are predominantly of IgG1 isotype (Figs. 2c and 3a). In contrast, plasmablasts of cluster 3 from patients #7 to #9 showed an enrichment of genes obtained from IL-2/IL-21-activated B cells treated for 24 and 48 h with IL-6/IL-21 and TGF-β, and express predominantly IgA1 and IgA2 (Figs. 2c and 3a). These gene sets include genes known to be induced by Stat3 and SMADs[27–34] (Supplementary Fig. 1c).

The termination of interferon-instruction of plasmablasts parallels the dynamics of serum concentrations of type II IFNs, which were prominent in the plasma at the early time points, but weaning later in the seven patients monitored (Fig. 3b). Conversely, concentrations of TGF-β were increasing with time (Fig. 3b). Accordingly, plasmablasts of patients beyond day 7 expressed the signature genes of IL2/IL21-activated B cells which had been further stimulated with additional TGF-β (Fig. 3a). In summary, severely affected COVID-19 patients show a continued immune reaction with egress of plasmablasts and plasma cells into the blood, which initially is controlled by IFNs, IL-2, and TGF-β, and later on predominantly by TGF-β.

**COVID-19 T cells express TGF-β.** The obvious involvement of IL-21 in the instruction of plasmablasts of COVID-19 patients points to cognate B cell activation by follicular helper T (Tfh)/Th17 cells, which are the main producers of IL-21[35]. Moreover, enhanced TGF-β expression has been described to be a hallmark of continued activation of Th17 cells[36,37]. TGF-β could also be provided by other sources, like subepithelial dendritic cells in Peyer's patches[38], regulatory T cells (Tregs) or neutrophils from the peripheral blood and airway tissue[39]. Interestingly, it has been reported that during SARS-CoV infection, cause of the SARS outbreak in 2003, TGF-β1 was increased in mucosal tissues[40]. Furthermore, SARS-CoV nucleoprotein (NP), which is 90% homologous to SARS-CoV-2 NP[41], can directly activate SMAD3, enhancing TGF-β-mediated gene expression[42], which could also include TGF-β itself[43]. Therefore, we focused our next analysis on SARS-CoV-2 reactive CD4+ T lymphocytes. We analyzed transcriptomes and TCR repertoires of single T cells from three patients who had been at ICU for 13, 29, and 32 days, respectively (Supplementary Table 2), and who were seropositive for IgM and IgG, and one of them also for IgA (Fig. 4a), focusing on their expression of *IL21* and *TGFB1*. To this end, PBMCs were stimulated for 6 h with a mixed peptide pool representing the SARS-CoV-2 spike glycoprotein (S), the membrane glycoprotein (M), and the nucleocapsid phosphoprotein (NP). Reactive CD4+ T lymphocytes expressing CD137 or CD154 were enriched

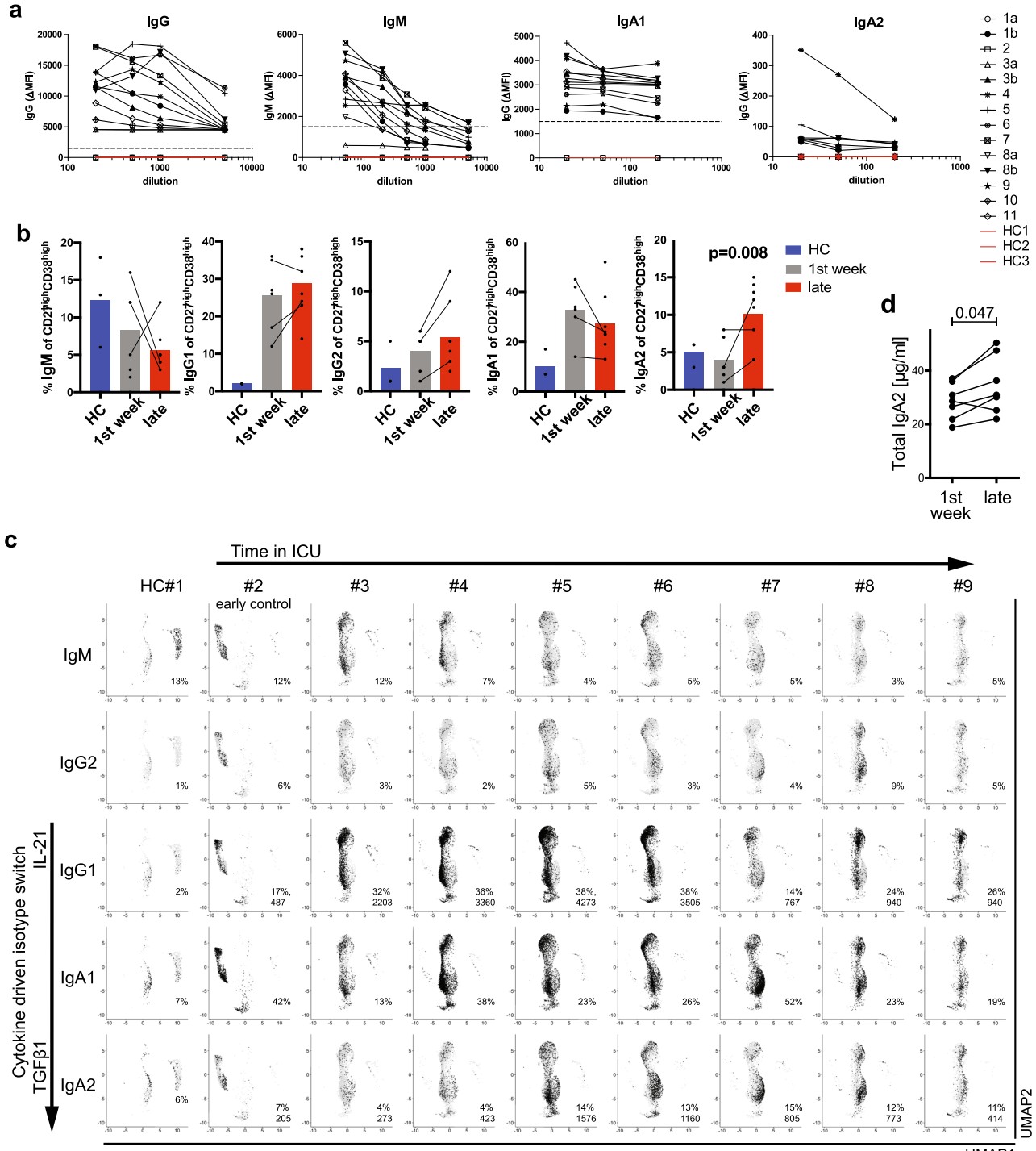

**Fig. 2 Peripheral plasmablasts switch to IgA2 in COVID-19 patients with prolonged stay in ICU. a** Binding of IgG, IgM, IgA1, and IgA2 antibodies to SARS-Cov-2-S protein expressed on the surface of transfected 293T cells. Data represent ΔMFI, which was calculated as following: MFI of transfected cells, incubated with sera and stained with respective anti-hIg, minus MFI of untransfected cells, incubated with sera and stained with respective anti-hIg. Dotted line indicates a ΔMFI of 1500. **b** Percentage of isotype-positive cells among sequenced CD27$^{high}$ CD38$^{high}$ B cells determined by single-cell BCR sequencing. Donors were grouped as "HC" for healthy controls (three donors), "1st week" for patients within 7 days after ICU admission (six donors) and "Late" for patients who have been admitted to the ICU for more than a week at the time of analysis (eight donors). Lines indicate measurements from the same patient at different time points. Two-sided Mann–Whitney *U* test. **c** UMAP representation of immunoglobulin isotype expression. Percentages refer to the total number of sequenced cells per donor. For IgG1 and IgA2 of COVID-19 patients the absolute number of cells is shown. Arrows represent the time (5–31 days and see Supplementary Table 1) patients had spent in the ICU at the time of analysis (upper) and the shift in cytokine milieu to allow switching to IgG1/IgA1/IgA2 (left). **d** Concentration of total IgA in the serum of COVID-19 patients at both early (less than a week) and late time points after ICU admission as determined by ELISA. Lines indicate measurements from the same patient at different time points. Two-sided Wilcoxon exact test.

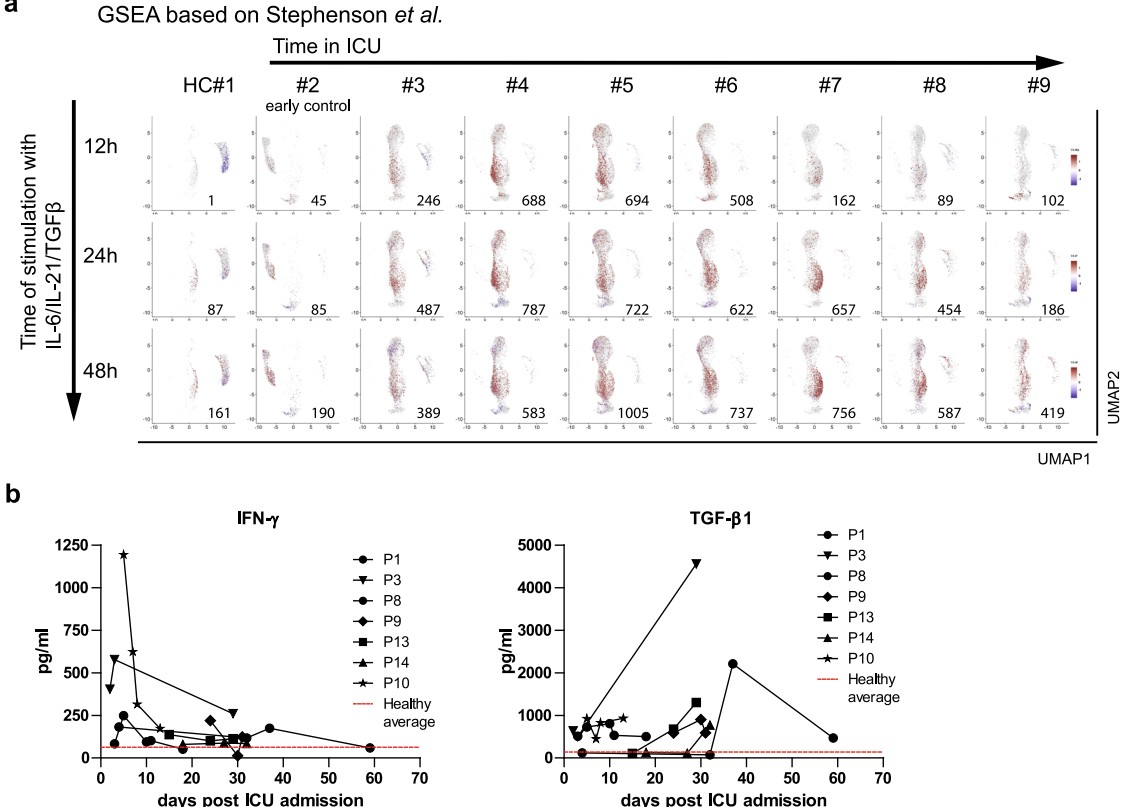

**Fig. 3 Peripheral plasmablasts of COVID-19 patients change their transcriptional signature in response to IL-21 and TGF-β in a kinetic fashion. a** Gene set enrichment analysis (GSEA) for the IL-6/IL-21/TGF-β signature as defined by contrasting IL-2/IL-21-activated B cells (plasmablasts) and IL-2/IL-21-activated B cells with an additional IL-6/IL-21/type I IFN, IL-6/IL-21/TGF-β or IL-6/IL-21/type I IFN/TGF-β treatment for 12, 24, and 48 h (stimulation data by Stephen et al.[26]). GSEA was performed for each cell relative to the mean expression in COVID-19 and healthy control samples (see Fig. 1). Cells were colored by the normalized enrichment score (NES) if statistically significant, with red and blue color shades for up-regulation and down-regulation. Shown numbers refer to the number of cells showing significant up-regulation of the gene set. Arrows represent the time (5–31 days and see Supplementary Table 1) patients had spent in the ICU at the time of analysis (upper) and the time cells were additionally stimulated to generate the data sets used (left). **b** Quantification of IFN-γ and TGF-β1 in the plasma of COVID-19 patients by cytokine bead arrays at several time points following ICU admission. Cytokine concentration in healthy individuals: IFN-γ, 63 ± 16 pg/mL; TGF-β1, 141 ± 74 pg/mL.

magnetically, representing antigen-experienced regulatory and effector cells, respectively (Fig. 4b, Supplementary Fig. 4)[44]. While we were able to isolate only very few of such cells from a healthy donor, frequencies of antigen-reactive CD4+ T cells among total CD4+ T cells in the three patients were 3.5%, 1.5%, and 4.9%, respectively (Supplementary Fig. 4). UMAP clustering identified two major populations expressing *CD3E*: Tregs expressing *FOXP3* and *IKZF2* (Helios) and Tfh cells expressing *ICOS* and *PDCD1* (Fig. 4c, d). After 6 h of antigenic stimulation Tfh cells expressed *CD40LG, IFNg, IL2, TNF*, but lacked the expression of *IL17, IL10* and all type 2 T helper cell-related cytokines (Fig. 4d). T helper (Th) cells expressing *IL21* (1–10%), *TGFB1* (2.9–9.5%), and *IL21* plus *TGFB1* (0.3–3.5%) were as frequent among the SARS-CoV-2-reactive Th cells as among measles-reactive Th cells of a healthy control (2%, 6%, 0.6%, Fig. 5). No significant numbers of *IL21* and/or *TGFB1* expressing SARS-CoV-2 reactive Th cells could be isolated from a healthy control (Fig. 5, Supplementary Fig. 4). Of note, the TCR repertoires of SARS-CoV-2-specific Tregs and Tfh cells are unique and not overlapping, with only 13 out of 1473 analyzed TCR clones being present in both subsets, a phenomenon that has also been observed for aero-antigen-specific Tregs and conventional T cells[44]. Taken together, these data demonstrate that severely affected COVID-19 patients have significant populations of circulating, SARS-CoV-2-experienced CD4+ T lymphocytes which are imprinted to express IL-21 and TGF-β,

and thus qualify as instructors of B cell activation in continued, SARS-CoV-2-triggered immune reactions.

**IgA plasmablasts of COVID-19 do not bind SARS-CoV-2 S or NP.** While it is obvious that the continued immune reactions of ICU COVID-19 patients are triggered by SARS-CoV-2, it is less clear whether they are also directed (exclusively) against SARS-CoV-2. The antigen-receptor repertoire of the plasmablasts is rather oligoclonal (Supplementary Fig. 5a) and shows somatic hypermutation, as demonstrated here for selected, expanded clonal families (Supplementary Fig. 5b), a hallmark of cognate interaction with CD4+ T cells[7]. Metadata on prominent clonal families of each patient are available as in Supplementary Data 1. Important is the fact that while all late ICU patients showed serum IgG, and one-third also IgA1 antibodies specific for the S protein of SARS-CoV-2, only one (patient #4) was seropositive for S-specific IgA2 (Fig. 2a), suggesting that the continued immune reaction to SARS-CoV-2 did not contribute significantly to humoral immunity against the virus. In line with this, only 3/9 seropositive patient samples (#8, #10, and #14) had detectable titers of neutralizing antibodies, as determined by a pseudotyped virus neutralization assay, late in ICU, despite the finding that all of the seropositive patient samples contained RBD-binding antibodies. Noteworthy, in our cohort the rate of patients in

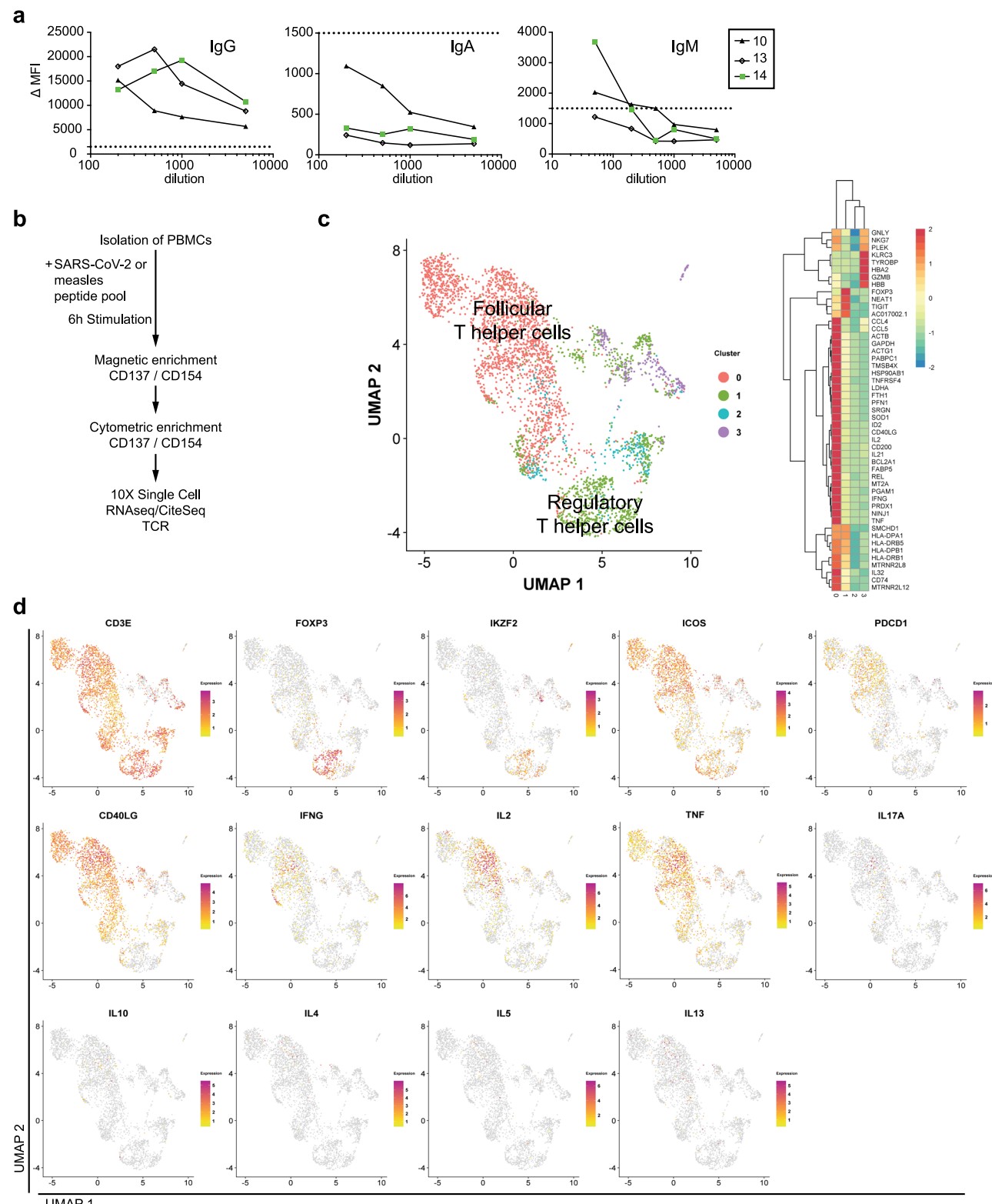

the ICU who do not develop detectable levels of neutralizing antibodies is higher than in other studies[45,46]. None of five cloned BCRs, representing five different expanded clones of plasmablasts (Supplementary Fig. 5b), recognize the SARS-CoV-2 S or NP. This finding is distinctly different from reported evidence for Influenza, where about 50% of the circulating activated B cells are hemagglutinin-specific[47]. In view of the finding that IgA-expressing plasmablasts in the chronic phase of severe COVID-

19 apparently mostly do not recognize SARS-CoV-2 S or NP proteins, it could be speculated, that if at all, they also may receive help from T cells with different specificities, yet to be identified.

**No IgA plasma cells in COVID-19 lungs**. If not relevant for systemic humoral immunity to SARS-CoV-2, the prolonged

**Fig. 4 SARS-CoV-2-reactive, antigen-experienced CD4⁺ T lymphocytes are in the circulation of COVID-19 ICU patients. a** Titration of SARS-Cov-2-S protein-specific IgG, IgA, and IgM antibodies in the serum of three COVID-19 ICU patients. ΔMFI calculation as described in Fig. 2a. Dotted line indicates a ΔMFI of 1500. **b** Workflow for the isolation and analysis of SARS-CoV-2-and measles-reactive CD4⁺ T lymphocytes. **c** Cells for single cell sequencing were sorted using FACS as PI⁻CD19⁻CD14⁻CD3⁺CD4⁺ CD154⁺ or CD137⁺ (gating strategy as used for the analysis shown in Supplementary Fig. 4). UMAP representation of 3727 CD4⁺ T cells from three COVID-19 ICU patients and two healthy controls (SARS-CoV-2-stimulated cells from one control and measles-stimulated cells from another control). Four clusters were identified based on transcriptional similarity using shared nearest-neighbor (SNN) modularity optimization (left). An additional CD14⁺ cluster (124 cells) and six outlying T cells with considerably less UMI-counts than the average were also identified but excluded from further analysis. Heatmap of signature genes of the four clusters. Depicted are genes with an absolute log2 fold change > log2(2) and a *p*-value < 0.01 after Bonferroni correction (two-sided Wilcoxon rank sum test). Colors corresponds to *z*-scores of the average expression. **d** Expression levels of selected signature genes and cytokines.

immune reaction and switch to IgA2 could contribute to local immunity at mucosal surfaces[48]. All analyzed plasmablasts and plasma cells expressed the J chain required to dimerize IgA and excrete it into the mucosal lumen (*IGJ*; Supplementary Fig. 1c)[49]. They also expressed CCR10 and ITGAE which could guide them into mucosal tissue (Supplementary Fig. 1c). It is therefore quite surprising, that in the lungs of three deceased COVID-19 patients (see Supplementary Table 3 for patient demographics and disease history), IgA or IgA2-expressing plasmablasts/plasma cells were rare, even when the patient had been affected by sepsis due to bacterial superinfection, while IgA, and in particular IgA2, -plasmablasts/plasma cells were prominent in the lungs of patients with COVID-19-unrelated pneumonia (Fig. 6a–c, Supplementary Fig. 6). In line with this observation, among cells isolated from broncheoalveolar lavage (BAL) fluid of patient #1, at day 59 (5459 cells), and patient #9, at days 31 (433 cells) and 46 (2723 cells) after ICU admission, only 42 cells in patient #1 on day 59, 10 cells in patient #9 on day 31 and 143 cells on day 46 were B cells (Fig. 7a, b, Supplementary Fig. 7a–c). The BAL was composed mainly of *CD14*-expressing cells and T cells, which expressed *CD40LG*, *IFNG*, and *TGFB1* and few of them *IL21* (Fig. 7a, b, Supplementary Fig. 7b, c). In patient #9 we observed an increase of cell infiltration on day 46 as compared to day 31. The majority of these cells expressed *TGFB1* (Fig. 7b).

In conclusion, we demonstrate here that severely affected COVID-19 patients which required prolonged ICU care show a continued immune reaction reflected by egress of plasmablasts into the blood. This immune reaction is initially controlled by IFNs, IL-21, and TGF-β, which target antibody class switching to IgG1 and IgA1. At later time points IFN is no longer involved, and the immune reactions are controlled by IL-21 and TGF-β, which in the end drives cells to switch to the terminal antibody class IgA2. Such cells do not relocate to the lung and they contribute little to humoral immunity to SARS-CoV-2. The specificities of the antibodies generated remain to be identified, but most of them are not specific for the spike protein, its receptor-binding domain (RBD) or NP. Whether or not the antibodies generated in the continued immune reactions of COVID-19 patients in the ICU are harmless, or whether they may even cause detrimental immunopathology remains to be shown. Therapeutic targeting of TGF-β may be a way to ameliorate severe COVID-19, especially, when considering the fibrosis-inducing capacity of TGF-β[50].

## Methods

**Human donors**. The recruitment of study subjects was conducted in accordance with the Ethics Committee of the Charité (EA 1/144/13, EA 1/075/19, EA 2/066/20, and EA 1/342/14) and was in compliance with the Declaration of Helsinki. Informed consent was obtained from all blood/brochoalveolar lavage donors included in the study. Four healthy adults (3 male and 1 female; average age 47 ± 7 (SEM)) and 13 COVID-19 patients (7 male and 6 female; average age 74 ± 11 (SEM)) with either documented disease history or exposure through natural SARS-CoV-2 infection that were verified by the levels of antigen-reactive antibody IgG titers.

**Sample collection and preparation**. Four to twenty milliliters of peripheral blood were drawn from each donor into Vacutainer® K2E (EDTA) Plus Blood Collection Tubes (BD Biosciences, Plymouth, UK). Blood samples were subjected to immediate preparation. In cases where cell enrichment was not performed directly from blood, mononuclear cells were isolated by density gradient centrifugation using Ficoll-Hypaque (Sigma-Aldrich). Serum samples were collected on the same time into Vacutainer® SST™ tubes (BD Biosciences, Plymouth, UK). Alternatively, the plasma fraction was isolated from EDTA blood. Bronchoalveolar lavage (BAL) was passed via 70 mcm cell strainer and lymphoid cells were isolated using anti-human CD45 microbeads (Miltenyi Biotec) and live cells were further sorted using a MA900 Multi-Application Cell Sorter (Sony Biotechnology). Both the serum and plasma samples were stored at −20 °C until further use.

**B cell isolation**. B cells were enriched from peripheral blood using StraightFrom Whole Blood CD19 MicroBeads (Miltenyi Biotec) according to manufacturer's instructions. Enriched cells were incubated with Fc Blocking Reagent (Milteniy Biotec) following manufacturer's instructions and subsequently stained up to 5 × 10⁶ cells per 100 μL with the following anti-human antibodies: CD3 (BW264/56, VioBlue, Miltenyi Biotec, Cat. No. 130-113-133, 1:400), CD14 (TÜK4, VioBlue, Miltenyi Biotec, Cat. No. 130-113-152, 1:200), CD16 (REA423, VioBlue, Miltenyi Biotec, Cat. No. 130-113-958, 1:100), CD27 (MT271, PE, Miltenyi Biotec, Cat. No. 130-097-926, 1:100), and CD38 (HIT2, APC, BioLegend, Cat. No. 303510, 1:20). DAPI was added before sorting to allow dead cell exclusion. Activated/differentiated B cells were identified and sorted as DAPI⁻CD3⁻CD14⁻CD16⁻CD38^high CD27^high. Alternatively, B cells were sorted from frozen PBMCs. To that end, PBMCs were stained as described but with the following anti-human antibodies: CD19 (LT19, Miltenyi Biotec, Cat. No. 130-113-728, 1:400), CD3 (BW264/56, VioBlue, Miltenyi Biotec, Cat. No. 130-113-133, 1:400), CD14 (TÜK4, VioBlue, Miltenyi Biotec, Cat. No. 130-113-152, 1:200), CD27 (MT271, PE, Miltenyi Biotec, Cat. No. 130-097-926, 1:100), and CD38 (IB6, PE-Vio770, Miltenyi Biotec, Cat. No. 130-113-990, 1:200). DAPI was added before sorting to allow dead cell exclusion. Activated/differentiated B cells were identified and sorted as DAPI⁻CD3⁻CD14⁻CD19⁺CD38^high CD27^high. All sortings were preformed using a MA900 Multi-Application Cell Sorter (Sony Biotechnology). Cell counting was performed using a MACSQuant flow cytometer (Miltenyi Biotec). The sorted CD38^high CD27^high cells were further processed for single cell RNA sequencing.

**T cell isolation and flow cytometric analysis**. Isolation of SARS-CoV-2reactive effector/memory and regulatory CD4⁺ T lymphocytes was performed as described (Okhrimenko, A, 2014, PNAS). At least 5 × 10⁶ PBMCs were stimulated for 6 h with with anti-CD28 (1 μg/mL) and the mixed spike glycoprotein (S), the membrane glycoprotein (M), and the nucleocapsid phosphoprotein (N) of the SARS-CoV-2, each 1 μg/peptide/mL, in the presence of 1 μg/mL anti-CD40. Stimulated cells were enrichment by MACS for CD137⁺ and CD154⁺ cells in two consecutive MS columns (Miltenyi Biotec) and the enriched cells incubated with anti-human Citeseq antibodies (CD154 Cat. No. 310849, CD127 Cat. No. 351356, CD45RA Cat. No. 310951, CD45RO Cat. No. 304163, CD69 Cat. No. 304259, CD39 Cat. No. 328237, HLA-DR Cat. No. 307663, CD279 Cat. No. 329963, CD57 Cat. No. 393321, CD27 Cat. No. 305651, and CD95 Cat. No. 302853, all from Biolegend and used according to manufacturer's instructions) for 30 min. Cells were also stained for 15 min with the following anti-human antibodies: CD3 (OKT3, BV785, Biolegend, Cat. No. 317330, 1:100), CD4 (SK3, PE-Cy5.5, Biolegend, Cat. No. 35-0047-42, 1:200), CD19 (H1B19, V500, BD Biosciences, Cat. No. 561121, 1:100), CD14 (TM1, Pacific Orange, DRFZ in-house conjugation, 1:100), CD137 (4B4-1, PE, Miltenyi Biotec, Cat. No. 130-093-476, 1:25), and CD154 (5C8, biotin, Miltenyi Biotec, Cat. No. 5190204135, 1:10). Streptavidin (eFluor450, eBioscience, Cat. No. 48-4317-1597, 1:200) was used subsequently. Total antigen-reactive effector/memory and regulatory CD4⁺ T cells were identified as PI⁻CD19⁻CD14⁻CD3⁺CD4⁺ CD154⁺ and/or CD137⁺ and sorted using a MA900 Multi-Application Cell Sorter (Sony Biotechnology). Cell counting was performed using a MACS-Quant flow cytometer (Miltenyi Biotec). The sorted SARS-CoV-2-reactive CD4⁺ T cells were further processed for single cell RNA sequencing. As a control measles-reactive memory CD4⁺ T lymphocytes were also isolated and further processed for single cell sequencing. To that end, PBMCs were stimulated as described, but in

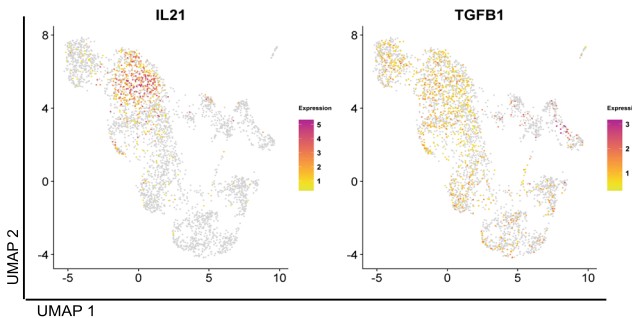

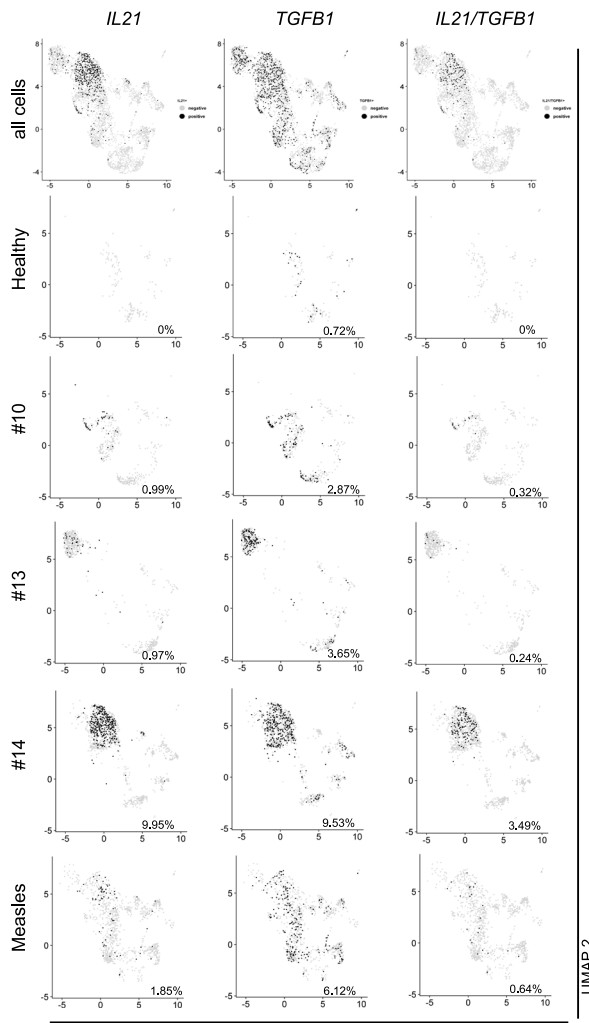

**Fig. 5 SARS-CoV-2-reactive, antigen-experienced CD4⁺ T lymphocytes of COVID-19 ICU patients express IL-21 and/or TFG-β.** Expression levels of *IL21* and *TGFB1* (top, color scale) in stimulated CD4⁺ T cells from three COVID-19 ICU patients and two healthy controls (see Fig. 4b, c). *IL21* single producers, *TGFB1* single producers, and *IL21/TGFB1* double producers among all sequenced cells, indicated for the individual donors (bottom, gray scale).The percentage of producer cells among total cells per donor is indicated in each UMAP.

presence of 5 µg/mL measles lysate instead of SARS-CoV-2 proteins. Measles-reactive memory CD4⁺ T cells were identified and sorted by the expression of PI⁻CD19⁻CD14⁻CD3⁺CD4⁺CD45RO⁺CD154⁺.

The phenotype of the SARS-CoV-2-stimulated cells was analyzed by flow cytometry, staining with the following anti-human antibodies: CD3 (OKT3, BV785, Biolegend Cat. No. 317330, 1:100), CD19 (H1B19, V500, BD Cat. No. 561121, 1:100), CD14 (TM1, Pacific Orange, in house conjugated, 1:100), CD4 (SK3, PE-Cy5.5, Biolegend Cat. No. 35-0047-42, 1:200), CD45RA (HI100, BV605,

Biolegend Cat. No. 304133, 1:200), CCR7 (G043H7, A488, Biolegend Cat. No. 353206, 1:100), CD69 (FN50, PE-CF594, BD Biosciences Cat. No. 5049599, 1:200), HLA-DR (L243, APC-Cy7, Biolegend Cat. No. 307617, 1:100), CD154 (24-31, BV421, Biolegend Cat. No. 310824, 1:100), CD137 (4B4-1, PE, Miltenyi Biotec Cat. No. 130-093-476, 1:25), IFNγ (4S.B3, PE-Cy7, Biolegend Cat. No. 502528, 1:200), IL-2 (MQ1-17H12, APC-Cy7, Biolegend Cat. No. 500342, 1:100), TNFα (Mab11, APC, BD Pharminogen Cat. No. 554514, 1:100), and fixable Live/Dead or propidium iodide (PI). Cells were acquired using a LSRFortessa flow cytometer (BD Biosciences) with FACSDiva (BD Biosciences) software and analyzed with FlowJo (Tree Star).

**Single cell RNA-library preparation and sequencing**. Single cell suspensions were obtained by cell sorting and applied to the 10x Genomics workflow for cell capturing and scRNA gene expression (GEX) and TCR/BCR/CiteSeq library preparation using the Chromium Single Cell 5′ Library & Gel Bead Kit as well as the Single Cell 5′ Feature Barcode Library Kit (10x Genomics). After cDNA amplification the CiteSeq libraries were prepared separately using the Single Index Kit N Set A. TCR/BCR target enrichment was performed using the Chromium Single Cell V(D)J Enrichment Kit for Human T cells and B cells, respectively. Final GEX and TCR/BCR libraries were obtained after fragmentation, adapter ligation, and final Index PCR using the Single Index Kit T Set A. Qubit HS DNA assay kit (Life Technologies) was used for library quantification and fragment sizes were determined using the Fragment Analyzer with the HS NGS Fragment Kit (1–6000 bp) (Agilent).

Sequencing was performed on a NextSeq500 device (Illumina) using High Output v2 Kits (150 cycles) with the recommended sequencing conditions for 5′ GEX libraries (read1: 26nt, read2: 98nt, index1: 8nt, index2: n.a.) and Mid Output v2 Kits (300 cycles) for TCR/BCR libraries (read1: 150nt, read2: 150nt, index1: 8nt, index2: n.a., 20% PhiX spike-in).

**Single-cell transcriptome and repertoire profiling**. Raw sequence reads were processed using cellranger (version 3.1.0), including the default detection of intact cells. Mkfastq, count and vdj were used in default parameter settings for demultiplexing, quantifying the gene expression and assembly of the B cell and T cell receptor sequences. Refdata-cellranger-hg19-1.2.0 and refdata-cellranger-vdj-GRCh38-alts-ensembl-2.0.0 were used as reference. The number of expected cells was set to 3000. Data from transcriptome sequencing and immune profiling is available in GEO under the accession GSE158038.

The cellranger output was further analyzed in R (version 3.6.1) using the Seurat package (version 3.1.1)[51]. In particular, transcriptome profiles for 13 COVID patients and 5 healthy controls were merged, normalized, variable genes were detected and a uniform manifold approximation and projection (UMAP) was performed in default parameter settings using FindVariableGenes, RunPCA and RunUMAP with 30 principle components. Expression values are represented as ln (10,000 * UMIsGene)/UMIsTotal + 1). Transcriptionally similar clusters were identified using shared nearest neighbor (SNN) modularity optimization, SNN resolutions ranging from 0.1 to 1.0 in 0.1 increments were computed, or gating was performed manually using the Loupe Browser (10x Genomics). Subsequently, clusters were annotated by projection of *CD19*, *MS4A1*, *CD27*, *CD38*, *IFIT1*, *MIKI67*, *IRF4* and *PRDM1* expression on the UMAP to assign different activation/proliferation/differentiation stages of B cell and signature genes were identified using FindAllMarkers in default parameter settings. Heatmaps are based on z-transformed expression values for genes with significant differences to means in different clusters as judged by a Bonferroni corrected P-value (Wilcoxon rank sum Test) below 0.01 and an minimal absolute fold-change to the mean of log2(1.3).

Data from transcriptome and immune profiling were merged by the same cellular barcodes. The high-confidence contig sequences for barcodes with known transcriptional profile were reanalyzed using HighV-QUEST at IMGT web portal for immunoglobulin (IMGT) to retrieve the V-, J- and D-genes as nucleotide and amino acid CDR3 sequence. IMGT-gapped-nt-sequences, V-REGION-mutation-and-AA-change-table as well as nt-mutation-statistics were used to determine the corresponding gapped germline FR1, CDR1, FR2, CDR2, FR3 sequences as well as estimated mutation counts in the FR1–FR3 region. The most abundant contig for the heavy and light BCR chains were assigned to the corresponding cell in the single cell transcriptome analysis. Cells with incomplete heavy and light chain annotation were removed from further analysis. This led to the annotation of 6258, 2678, 2437, 7480, 4704, 7527, 4922, 4743, and 9168 transcriptome profiles for 9 COVID Patients and 1571, 1621, and 2633 for three healthy donors. In case of two contigs for the heavy chain the alternative contig was defined as the inactive heavy chain gene locus transcript, if found nonproductive. For the hypermutation analysis clonal families were defined by the same VJ-gene usage, gapped germline FR1–FR3 sequence and the nucleotide CDR3 sequence length of the heavy and light chains. The hypermutation trees were computed using GLaMST with concatenated FR1–FR3 sequences of the heavy and light chain and the germline sequence as root input. The effective diversity for different Hill orders were computed based on the FR1–FR3 germline sequence, used vj-gene and the cdr3nt sequence. The same procedure was used for the analysis of time changes between different time points for patients #1 and #9. The first samples at day 4 and day 31 were reanalyzed with the corresponding samples at day 59 and day 46 for patient

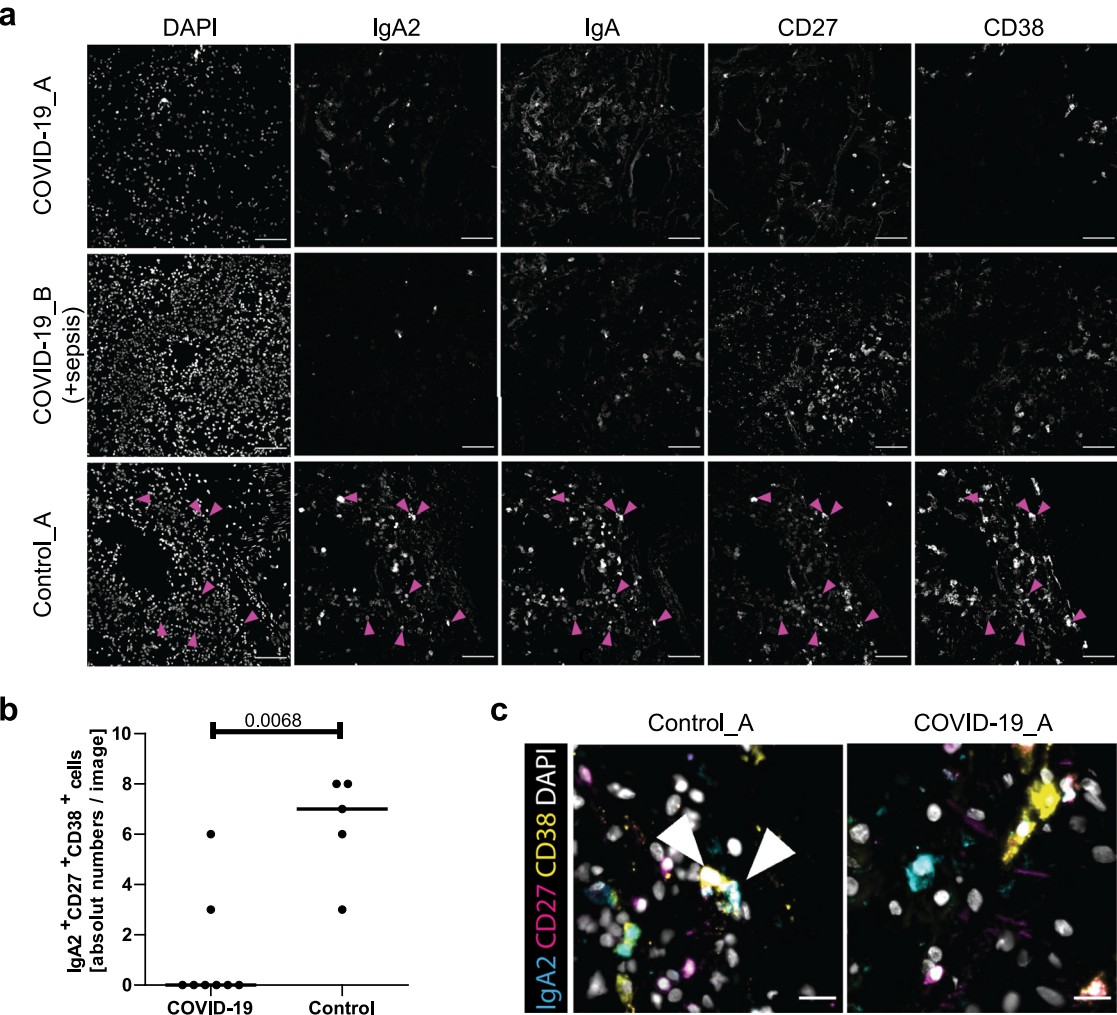

**Fig. 6 IgA-expressing plasmablasts and/or plasma cells are absent from the lungs of deceased COVID-19 patients.** The presence of plasmablasts in the lungs of SARS-CoV-2 patients and controls (see Supplementary Table 3 for donor information) was examined by MELC by analysisng two fields of view per patient, except for COVID-19_A with four fields of view analyzed and Control_B with one single field of view analyzed. Images are representative of these fields of view. **a** 5-marker MELC panel of SARS-CoV2-positive and control lungs (SARS-CoV2-negative). Each image of each patient depicts the same field of view of the same section, sequentially stained with the fluorescence-labeled antibodies indicated and the nuclear stain DAPI. Magenta arrows indicate IgA2+IgA+CD27+CD38+ cells (containing a nucleus). Images contain 2048 × 2048 pixels and are generated using an inverted wide-field fluorescence microscope with a ×20 objective, a lateral resolution of 325 nm and an axial resolution above 5 μm. Scale bar: 100 μm. **b** Absolute numbers of IgA2+IgA+CD27+CD38+ cells per field of view in all MELC runs acquired . Each field of view is represented by a circle. The line indicates the median. Unpaired two sided Mann–Whitney $U$ test. **c** Region of interest of one exemplary control and COVID-19 lung (as in **a**), showing an overlay of the indicated markers. White arrows point out IgA2+CD27+CD38+ cells. Scale bar: 20 μm.

#1 and patient #9, respectively. For the overlaps between BAL and peripheral blood sample of patient #1, a separated immune profile was used.

For TCR analysis samples from three COVID-19 patients and two healthy control were tagged with citeseq antibodies, pooled, sequenced, and analyzed in analogy to the B cells. Transcriptome and immune profiles were merged by same barcodes. For the V, D, and J, the CDR3nt, CDR3aa, and Isotype were directly taken from the cellranger results. In case of more than one contig for the heavy or light TCR chain the most abundant, productive contig was chosen. Different sample origin was demultiplexed using sample-specific hash code.

**Gene set enrichment analysis (GSEA)**. The IL2/IL21 signature of activated B cells/plasmablasts and additional subsequent treatment with IL6/IL21/IFN I, IL6/IL21/TGF-β, and IL6/IL21/IFN I/TGF-β was derived from a publication of Stephenson and colleagues[26] (https://doi.org/10.4049/jimmunol.1801407). Quantile normalized Illumina HumanHT-12 beadchip expression values for Il2/IL21-activated B cells with and without additional treatment was obtained from GEO (accession number GSE120367). Sparse partial least-squares discriminant analysis (sPLS-DA) was performed using mixOmics R-package by keeping 200 genes in three components. Expression values from Stephenson et al. and the 10xGenomics data from this study was based on gene symbols. Solely genes with non-redundant symbols and unique equivalent in 10x genomics sequencing were used. sPLS-DA

was performed separately for 12, 24, and 48 h after stimulation. IL2/IL21-activated B cells at day 6 were used as control (Supplementary Fig. 8). *TGFB*-Signature genes were defined by positive loadings on component 2. Signature genes were defined by negative contribution to Component 1 for IL2/IL21-activated B cell signature, negative contribution to Component 2 for IL6/IL21/IFN I treatment, positive contribution to Component 3 for IL6/IL21/IFN I/TGF-β treatment, and positive contribution to Component 2 for IL6/IL21/TGF-β.

GSEA was performed for each cell, using the natural logarithm of the fold change expression relative to mean expression of all cell as pre ranked list and 1000 randomizations. Statistically significant upregulation or downregulation was defined by a FDR ≤ 0.25 and normalize $p$ value < 0.05[52]. For visualization NES for significant cells were plotted.

**Flow cytometric analysis of serum antibody titers**. HEK293T cells were transfected with a plasmid-expressing wild-type SARS-CoV-2 S protein[53]. Next day, transfected cells were collected and incubated with sera or recombinant anti-SARS-CoV-2 Spike Glycoprotein S1 antibody (CR3022, Abcam, Cat. No. ab273073, 1 μg/mL) for 30 min, washed twice with PBS/BSA and stained with anti-human IgG-Alexa647 (polyclonal, Southern Biotech Cat. No. 2014-31, 1:400), PE anti-IgM (CH2, Invitrogen Cat. No. MA1-10381, 1:200) and FITC anti-IgA (polyclonal, Southern Biotech Cat. No. 2052-02, 1:200), anti-human IgA1 Alexa488 (B3506B4, Southern Biotech Cat. No.

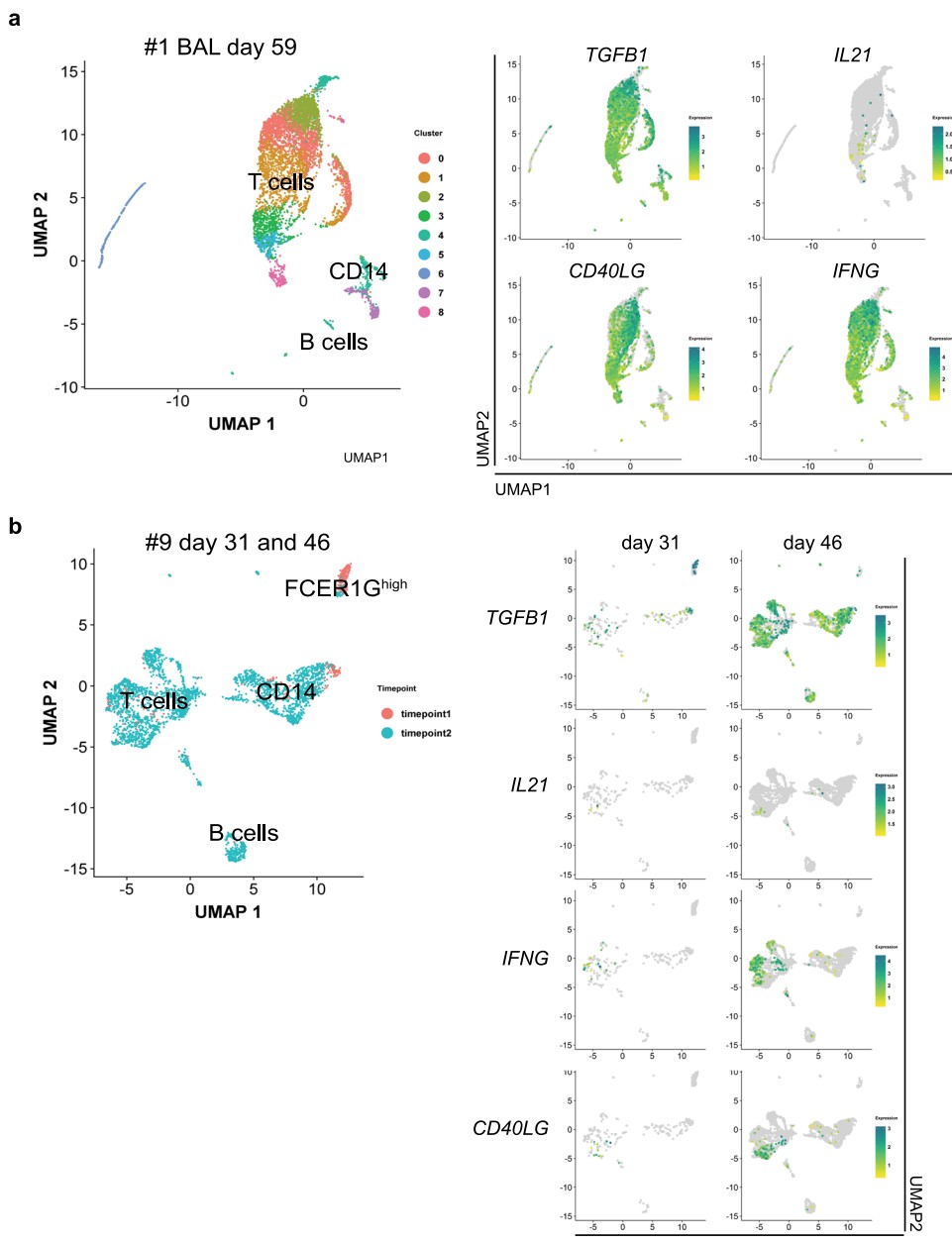

**Fig. 7 Cells isolated from brochoalveolar lavage of severe COVID-19 patients are enriched in TGFB1-expressing T and CD14-positive cells.**
Bronchoalveolar lavage (BAL) cells for single cell sequencing were enriched for CD45+ cells via MACS and live cells were further sorted using FACS (gating strategy in Supplementary Fig. 7a). **a** UMAP of 5459 cells representing clusters containing T cells, B cells, and *CD14*-expressing cells from patient #1 on day 59 following ICU admission. B and T cell cluster identification based on BCR/TCR, *CD19*, *CD3E*, *CD4*, and *CD8A* expression (see Supplementary Fig. 7b). UMAP representation of expression of *TGFB1*, *IL21*, *CD40LG*, and *IFNG*. **b** UMAP coordinates and clustering was computed for 433 and 2723 cells from patient #9 on days 31 and 46 following ICU admission, respectively. UMAP representation of expression of *TGFB1*, *IL21*, *CD40LG*, and *IFNG* at the two different time points is shown side by side.

9130-30, 1:200), anti-human IgA2-Alexa647 (A9604D2, Southern Biotech Cat. No. 9140-31, 1:200) for 30 min. Cells were washed with PBS/BSA and DAPI was added before analysis for dead cell exclusion. Samples were analyzed on a FACSCanto (BD Biosciences) and using FlowJo v10 (Tree Star Inc.) analysis software. Mean fluorescent intensities were determined for transfected and non-transfected cells of each sample. Data were presented as ΔMFI = MFI(transfected)−MFI(non-transfected).

**ELISA analysis of serum antibody titers**. Antibody titers against various Sars-Cov-2 S protein were measured using COVID-19 Human IgM IgG Assay Kit (Abnova, Cat. No. ABN-KA5826) with slight modifications: anti-S IgG and IgM were quantified according to manufacturer's instructions. To analyze S protein-specific IgA1 and IgA2 responses, detection was performed using mouse anti-human IgA1-AP (B3506B4, Southern Biotech Cat. No. 9130-04, 1:2000) and anti-

human IgA2-AP (A9604D2, Southern Biotech Cat. No. 9140-04, 1:2000), followed by pNPP substrate. Quantification of anti-S-RBD and anti-NP Sars-CoV-2 antibody responses has been performed using respective recombinant proteins. Briefly, plates were coated with 100 μL PBS containing 1 μg/ml of Spike RBD-His Recombinant Protein (Sino Biological) or SARS-CoV-2 Nucleocapsid His Protein (RnD systems) overnight at 4 °C. Plates were blocked with 5% PBS/BSA for one hour, serial dilution of sera were applied and incubated overnight at 4 °C. Antibodies were detected using anti-human IgG (polyclonal, MP biomedicals Cat. No. 59289, 1:2000), IgM (polyclonal, Sigma-Aldrich Cat. No. A3437-.25 ML, 1:1000), IgA1 (B3506B4, Southern Biotech Cat. No. 9130-04, 1:2000) and IgA2 (A9604D2, Southern Biotech Cat. No. 9140-04, 1:2000) coupled with alkaline phosphatase, followed by pNPP substrate. Reactions were stopped by adding NaOH and absorbance values at 405 nm were measured using SpectraMax plate reader (Molecular devices). Antibody titers were quantified using non-linear curve fit in GraphpadPrism 5.0.

Spearman rank correlation was applied for analysis of correlations between ELISA-based and flow cytometry-based antibody measurements.

**SARS-CoV-2 neutralization assay.** Pseudovirus neutralization assay for SARS-CoV-2 was preformed as previously described[54,55]. Briefly, 2 µL of serum or serum dilutions were pre-incubated for 1 h at 37 °C with 10 µL of virus in 10 µL PBS. The pre-incubated serum/virus suspensions were then added to VeroE6 cells cultured in 100 µL DMEM 5% hiFCS on a 96-well plate, and incubated at 37 °C for 1 day before readout (plaques/Fluorescent foci). Assay was performed in duplicate wells, three dilutions per serum. Analysis was done using averages of four fields of view chosen randomly (blind) and then assessed for foci using fluorescent microscopes.

**Antibody generation.** From the VH and VL sequences obtained by scRNA, synthetic GeneBlocks were synthesized (IDTDNA, Leuwen, Belgium) and cloned into expression vectors for human IgG1 and human Igk, essentially as described by Tiller et al.[56], except that we used Gibson assembly for the cloning. Heavy and light chain plasmids were transfected into 293 freestyle cells as recommended by the manufacturer and antibodies were purified on a Protein G column from the culture supernatants.

**Cytokine measurements.** Blood plasma was diluted 1:3, and IL-21, IFN-α, and IFN-γ were detected using a bead-based multiplex cytokine array (ProcartaPlex Human Cytokine Panel 1B, Thermo Fisher Scientific). TGF-β was detected using the ProcartaPlex Human TGF-beta 1 Simplex Kit (Thermo Fisher Scientific). Prior to measuring plasma-TGF-β1, the bioactive form of TGF-β1 was generated by incubating the plasma with 1 N HCl followed by neutralization with 1.2 N NaOH according to the manufacturer's instructions. All cytokines were measured using the Luminex MAGPIX instrument and quantified using the xPONENT analysis software (Luminex Corporation).

**Tissue preparation for MELC.** Fresh frozen lungs tissue was cut into 5 µm sections with a NX80 cryotome (ThermoFisher, Waltham, MA, USA), and deposited on APES-coated cover slides (24 × 60 mm; Menzel-Gläser, Braunschweig, Germany). Samples were fixed for 10 min at RT with 2% paraformaldehyde (methanol-free and RNAse-free; Electron Microscopy Sciences, Hatfield, Philadelphia, USA). After washing samples were permeabilized with 0.2% Triton X-100 in PBS for 10 min at room temperature and blocked with 10% goat serum and 1% BSA in PBS for at least 20 min. Afterwards, a fluid chamber holding 100 µl of PBS was created using "press-to-seal" silicone sheets (Life technologies, Carlsbad, CA, USA; 1.0 mm thickness), which were attached to the cover slip, surrounding the sample.

**MELC image acquisition.** We generated the multiplexed histology data on a modified Toponome Image Cycler® MM3 (TIC) originally produced by MelTec GmbH & Co.KG Magdeburg, Germany[57,58]. The robotic microscopic system consists of: (i) an inverted widefield (epi)fluorescence microscope Leica DM IRE2 equipped with a CMOS camera and a motor-controlled XY-stage, (ii) CAVRO XL3000 Pipette/Diluter (Tecan GmbH, Crailsheim, Germany), and (iii) a software MelTec TIC-Control for controlling microscope and pipetting system and for synchronized image acquisition. The MELC run is a sequence of cycles, each containing the following four steps: (i) incubation of the fluorescence-coupled antibody and subsequent washing; (ii) cross-correlation-based auto-focusing; (iii) photo-bleaching of the fluorophore; and (iv) a second autofocusing step followed by acquisition of a 3D stack post-bleaching fluorescence image. In each four-step cycle another fluorescence-labeled antibody is used. After the sample was labeled sequentially by DAPI (Roche, Cat. No. 10236276001, dilution 1:5000) and all antibodies of interest as described above, the experiment was completed. Fluorescence-coupled antibodies used: IgA2-PE (Clone REA995, Miltenyi, Cat. No. 130-117-763, dilution 1:50); CD27-PE (Clone REA499, Miltenyi, Cat. No. 130-114-166, dilution 1:50); CD38-PE (Clone IB6, Miltenyi, Cat. No. 130-113-427, dilution 1:50); and IgA-PE (Clone IS11-8E10, Miltenyi, Cat. No. 130-114-002, dilution 1:50). The antibodies were stained in the indicated order.

**Image pre-processing.** In short, all images were aligned based on the reference phase contrast image taken at the beginning of the measurement. Afterwards, each fluorescence MELC image was processed by background subtraction and illumination correction, based on the bleaching images[57]. In order to account for slice thickness, an "Extended Depth of Field" algorithm was applied on the 3D fluorescence stack in each cycle[59]. Images were then normalized in ImageJ[60], where a rolling ball algorithm was used for background estimation, edges were removed (accounting for the maximum allowed shift during the autofocus procedure) and fluorescence intensities were stretched to the full intensity range (16 bit → 2[16]).

**Autopsy tissue.** Autopsies were performed at the Department of Pathology and Neuropathology, Charité-Universitätsmedizin Berlin. For sampling postmortem tissue from COVID-19 and control patients, its virological assessment and the histological analysis, the approval from the Ethics Committee of the Charité included EA 1/144/13 with EA 1/075/19 and EA2/066/20 Substudy No. 60. The study was in compliance with the Declaration of Helsinki. In all included deceased patients a whole-body autopsy was performed, which included a thorough histopathologic and molecular

evaluation comprising virological assessment of SARS-CoV-2 RNA levels as previously described[61]. Clinical records were assessed for pre-existing medical conditions and medications, current medical course, and ante-mortem diagnostic findings as specified in Supplementary Table 3. Cause of death for the control cases was aspiration pneumonia and COVID-19 for the SARS-CoV-2-infected patients.

**Reporting summary.** Further information on research design is available in the Nature Research Reporting Summary linked to this article.

## Data availability

Next Generation Sequencing data sets generated and analyzed during the current study are available in the GEO repository under accession number GSE158038. Data used for GSEA was generated by Stephenson and colleagues and is publicly available under https://doi.org/10.4049/jimmunol.1801407. Source data are provided with this paper. All other data that support the findings of this study are available on request from the corresponding author [M.-F.M.]. Source data are provided with this paper.

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

## Acknowledgements

We thank Jenny Kirsch, Ana Catalina Teichmüller, and Toralf Kaiser for support in cell sorting. We are indebted to Francisca Egelhofer and Anistan Sebastiampillai for excellent technical assistance and advice. The authors are most grateful to the patients and their relatives for consenting to autopsy and subsequent research, which were done with support of the BrainBank of the Department of Neuropathology–Universitätsmedizin Berlin, Germany. This Work was supported by the state of Berlin and the "European Regional Development Fund" to M.-F.M. (ERDF 2014–2020, EFRE 1.8/11, Deutsches Rheuma-Forschungszentrum), the Berlin Institute of Health with the Starting Grant-Multi-Omics Characterization of SARS-CoV-2 infection, Project 6 "Identifying immunological targets in Covid-19" to H.-D.V., A.D., and M.-F.M., by the Deutsche Forschungsgemeinschaft (DFG) through the TRR130-P16 and TRR241-B03 to A.R. and H.-D.C. and TRR130-P17 to H.R., by the European Research Council through the Advanced Grant IMMEMO (ERC-2010-AdG.20100317 Grant 268978) to A.R., and by the DFG–Projektnummer 389687267 to J.D. and A.R. F.M. was supported by the Leibniz Association (Leibniz Collaborative Excellence, CHROQ K121-2018). A.K. was supported by the DFG (TRR241-A04), Russian Ministry of Science and Higher Education of the Russian Federation (grant 075-15-2019-1660) and Russian foundation for basic research (#17-00-00435). A.E.H. was funded by the DFG (TRR130-C01 and P17, HA5354/6-2 and HA5354/8-2). L.B., J.N., and H.-D.C. were funded by the Dr. Rolf M. Schwiete Foundation. T.H.W. was funded by the Deutsche Forschungsgemeinschaft-Projekt-ID 324392634-TRR221 and German Federal Ministry of Education and Research (BMBF)-COVID19-014 01KI2043A CoVER-Ab.

## Author contributions

Conceptualization: M.-F.M, A.R., F.M.; Methodology: M.F.-G., A.K., P.D., G.A.H., A.P.-R., W.D., S.F., K.H.; Software: F.H., P.D.; Validation: M.-F.M., M.F.-G., A.K., C.T., G.A.H.; Formal analysis: F.H., P.D.; Investigation: M.F.-G., A.K., C.T., G.A.H., A.P.-R., W.D., R.M., C.F., S.F., K.H., L.B., J.N., P.K.J., G.M.G., K.L., L.O., L.H., S.B.; Resources: G.S., M.R., T.K., M.A.M., S.A., S.T., V.M.C., S.E., H.R., M.W.; Data curation: M.-F.M, F.H., P.D.; Writing—original draft: A.R., F.M., M.-F.M., M.F.-G.; Writing—Review & editing: M.F.-G., G.A.H.; Visualization: F.H., P.D., A.K., M.-F.M, A.P.-R.; Supervision: H.-D.C., T.D., A.D., M.M., H.-D.V., T.H.W., J.D., A.E.H., H.R., M.W., F.M., A.R., M.-F.M.; Project administration: M.-F.M, A.R., F.M.; Funding acquisition: A.R., M.-F.M.

## Funding

## Competing interests

The authors declare no competing interests.

## Additional information

[1]Deutsches Rheuma-Forschungszentrum (DRFZ), an Institute of the Leibniz Association, Berlin, Germany. [2]Belozersky Institute of Physico-Chemical Biology and Biological Faculty, M.V. Lomonosov Moscow State University, Moscow, Russia. [3]Center for Precision Genome Editing and Genetic Technologies for Biomedicine, Engelhardt Institute of Molecular Biology, Russian Academy of Sciences, Moscow, Russia. [4]Laboratory of Innate Immunity, Department of Microbiology and Infection Immunology, Charité-Universitätsmedizin Berlin, Berlin, Germany. [5]Berlin Institute of Health (BIH), Berlin, Germany. [6]Mucosal and Developmental Immunology, Deutsches Rheuma-Forschungszentrum, Berlin, Germany. [7]Department of Rheumatology and Clinical Immunology, Charité-Universitätsmedizin Berlin, corporate member of Freie Universität Berlin, Humboldt-Universität zu Berlin, and Berlin Institute of Health, Berlin, Germany. [8]Department of Neuropathology, Charité-Universitätsmedizin Berlin, corporate member of Freie Universität Berlin, Humboldt-Universität zu Berlin, and Berlin Institute of Health, Berlin, Germany. [9]Dermatological Allergology, Department of Dermatology and Allergy, Charité-Universitätsmedizin Berlin, corporate member of Freie Universität Berlin, Humboldt-Universität zu Berlin, and Berlin Institute of Health, Berlin, Germany. [10]Friedrich-Alexander-University Erlangen-Nuremberg, Erlangen, Germany. [11]BIH Center for Regenerative Therapies (BCRT), Charité Universitätsmedizin Berlin, Berlin, Germany. [12]Technische Universität Berlin, Institute of Biotechnology, Berlin, Germany. [13]Institute of Virology, Charité-Universitätsmedizin Berlin, corporate member of Freie Universität Berlin, Humboldt-Universität zu Berlin, and Berlin Institute of Health, Berlin, Germany. [14]Department of Pediatric Pulmonology, Immunology and Critical Care Medicine, Charité-Universitätsmedizin Berlin, corporate member of Freie Universität Berlin, Humboldt-Universität zu Berlin, and Berlin Institute of Health, Berlin, Germany. [15]German Centre for Lung Research (DZL), associated partner site, Berlin, Germany. [16]Department of Anesthesiology and Intensive Care Medicine, Campus Benjamin Franklin, Charité - Universitätsmedizin Berlin, Corporate Member of Freie Universität Berlin, Humboldt-Universität zu Berlin, and Berlin Institute of Health, Berlin, Germany. [17]Institute of Medical Immunology, Charité - Universitätsmedizin Berlin, Corporate Member of Freie Universität Berlin, Humboldt-Universität zu Berlin, and Berlin Institute of Health, Berlin, Germany. [18]Institute of Pathology, Charité - Universitätsmedizin Berlin, Corporate Member of Freie Universität Berlin, Humboldt-Universität zu Berlin, and Berlin Institute of Health, Berlin, Germany. [19]These authors contributed equally: Marta Ferreira-Gomes, Andrey Kruglov, Pawel Durek, Frederik Heinrich, Caroline Tizian, Gitta Anne Heinz. [20]These authors jointly supervised this work: Helena Radbruch, Mario Witkowski, Fritz Melchers, Andreas Radbruch, Mir-Farzin Mashreghi. ✉email: mashreghi@drfz.de

