## [Peer Review File · Nature Communications]

REVIEWERS' COMMENTS

Reviewer #2 (Remarks to the Author):

The authors have addressed my comments. I have no further comments to make.

Reviewer #4 (Remarks to the Author):

I have read with interest the manuscript of Ferreira-Gomes and colleagues, as well as the point-by-point response. It appears that the concerns of predominantly reviewer #1 centred on the limited number of early ICU COVID patients analysed and the validity of the antibody data. The authors have addressed the first issue thoroughly by providing additional patient datasets for this period, as well as now providing 2 additional patients with longitudinal samples. The authors also provide additional serological data using a binding assay and conventional ELISA. Comparison of these techniques in Supplemental figure 2E suggests a strong correlation between the techniques.

In my opinion, these changes, along with extensive changes in the text make this manuscript suitable for publication in nat comms.

I would like to make one additional point for the authors to consider, although I stress this should not hold up the publication process.

Given that the cluster 2,3,4,5 in figure 1 are MS4A1-CD27+CD38+PRDM1+IRF4+IgJ+ and described figure 2 to be high Ig expressors, I wonder why the authors designate these cells to be "activated B cells". Typically in 10x data, B cells differentiation states cluster together and are very distinct from plasmablast/plasma cells. Given in figure 1 that clusters 2-5 are closer to the cluster 6 (labelled as plasma cells), then to the cluster 1, isn't it more likely that these cells are actually all plasmablasts? Are other PB/PC marker expressed in these cells, eg XBP1, TNFRSF17, SLAMF7, SDC1 etc.

Stephen Nutt

Reviewer #5 (Remarks to the Author):

My concerns with the manuscript has been addressed.

Point-by-Point response to the reviewer's comments on the revised version:

From the comments of the reviewers, we conclude that all reviewers are now satisfied with our revision. Reviewer #4 made an additional point that we considered and agreed with the proposed changes.

Reviewer #4:

I would like to make one additional point for the authors to consider, although I stress this should not hold up the publication process.

Given that the cluster 2,3,4,5 in figure 1 are MS4A1-CD27+CD38+PRDM1+IRF4+IgJ+ and described figure 2 to be high Ig expressors, I wonder why the authors designate these cells to be "activated B cells". Typically in 10x data, B cells differentiation states cluster together and are very distinct from plasmablast/plasma cells. Given in figure 1 that clusters 2-5 are closer to the cluster 6 (labelled as plasma cells), then to the cluster 1, isn't it more likely that these cells are actually all plasmablasts? Are other PB/PC marker expressed in these cells, eg XBP1, TNFRSF17, SLAMF7, SDC1 etc.

Our response: Our choice to designate clusters 2,3,4 and 5 as "activated B cells" came from the fact that these cells are somewhat heterogeneous. However, we agree that "plasmablasts" is a more accurate designation as these cells indeed express XBP1, TNFRSF17, SLAM7 and SDC1. We thank the reviewer for the recommendation.